# Tomato Fruit Quality as Affected by Ergonomic Conditions While Manually Harvested

**Łukasz Kuta** [1,*]**, Piotr Komarnicki** [2,*] **, Katarzyna Łakoma** [2] **and Joanna Praska** [1]

1   Institute of Environmental Protection and Development, The Faculty of Environmental Engineering and Geodesy, Wrocław University of Environmental and Life Sciences, 50-375 Wrocław, Poland; 121214@student.upwr.edu.pl
2   Institute of Agriculture Engineering, The Faculty of Life Sciences and Technology, Wrocław University of Environmental and Life Sciences, 50-375 Wrocław, Poland; 116155@student.upwr.edu.pl
*   Correspondence: lukasz.kuta@upwr.edu.pl (Ł.K.); piotr.komarnicki@upwr.edu.pl (P.K.)

**Abstract:** The harvest phase plays an important role in the whole process of production of tomato fruit. Therefore, it is necessary to ensure a technological process that will not damage biological materials. The harvest phase plays an important role in the whole process. Many growers use special machines for harvesting, but there are fruits and vegetables that should be harvested manually to avoid damaging the surface or parenchyma tissue of the harvested objects. In addition to maintaining the quality of biological materials, work comfort, and ergonomic conditions for pickers should be ensured because inadequate working conditions do not encourage employees to undertake manual work in horticulture. Therefore, there have been shortages of workers on Polish plantations in the past year. Based on manual tomato harvesting, the authors conducted a matched qualitative research study on biological materials and work ergonomics. For this purpose, the Grip System was used to investigate tomato quality by assessing the impact of picking hand pressure (in three different picker's body positions) on the harvested objects. Simultaneously, for the picker's ergonomic analysis, a non-invasive surface electromyography method was used to precisely measure changes in muscle motor unit action in the picker's wrist and lumbar spine while in three characteristic picker's positions. The tests found that the poorest body position was when the body was deeply inclined and simultaneously twisted. No significant effect was shown of the body position of the tomato picker on the deterioration of the picked fruit quality. However, body positions significantly affect the level of physical load and work comfort.

**Keywords:** tomato quality; harvest; surface pressure; muscle tension; EMG

## 1. Introduction

Tomatoes are the most popular fruit in the global market, with continually increasing consumption. As market demands rise, maintaining the highest level of tomato production quality is imperative [1,2]. Manual harvesting of tomatoes is labor-intensive and often uncomfortable for workers. Tomatoes are delicate and prone to bruising, making them unsuitable for automated harvesting systems [3,4]. A critical issue during harvesting is the gentle separation of the tomato from the stem. This separation is often overlooked when designing tomato grasping tools but is paramount in the manual harvesting process [5]. In traditional manual harvesting, pickers use their hands to delicately separate the fruit from the plant, employing techniques such as ripping, bending, twisting, and sometimes using cutting tools. Efficient manual harvesting requires skill and experience, while robotic harvesting often results in plant damage and poor performance. Developing a practical robotic harvesting solution is challenging due to the unique structure of fruit growth and development under real-world conditions [6]. In comparison to manual picking, automated picking is more likely to cause damage to tomatoes [7,8]. Human hands and bodies possess innate grasping abilities, with the sense of touch and muscular strength

enabling rapid adaptation to different crop shapes and textures to apply the appropriate force for separation [9]. However, human abilities are limited by fatigue. Given the high repetition of manual labor, it is essential from an ergonomics perspective to understand the musculoskeletal loads and forces involved when a picker's hand comes into contact with tomato fruit. Furthermore, in terms of quality preservation, knowledge of the mechanical properties of the fruit is crucial. Obtained data are valuable inputs for models that predict product quality and behavior before, during, and after harvest [10]. Firmness and resistance to compression are important parameters for describing tomato quality [11,12]. These parameters are linked to the maturity rate and susceptibility of tomatoes to damage at harvest. Understanding the contact pressures of hand fingers on the fruit allows for the prediction of potential damage levels and optimization of manual picking techniques [13].

Młotek et al. [14] demonstrated that when picking apples using the rotation technique, the index finger, along with the thumb, plays the most significant role in load transfer. There is a risk of causing permanent mechanical damage to the fruit, which can occur within an average pressure range of 0.1 to 0.2 MPa. One of the key global public health issues is the problem of musculoskeletal disorders (MDS) among workers, particularly during harvesting activities, whether static or dynamic in nature [15]. This issue also affects workers in horticulture and agriculture, as it is linked to physical strain and non-ergonomic body postures during daily work, resulting in significant ergonomic hazards in the workplace [16]. Harvest technique has a critical impact not only on the quality of tomatoes but also on the ergonomics of the picker's work. Tomato harvesting often involves several hours of work per day, leading to discomfort for pickers, characterized by muscle pain and fatigue in the upper limbs and spine [17]. Thetkathuek et al. [18] emphasized the importance of musculoskeletal diseases by comparing them to a global problem. This issue was also highlighted by Kamaruzaman and Fauzi [19], who described the ergonomic risks associated with upper limb work during repetitive activities, a scenario commonly encountered during fruit and vegetable harvesting. Prairie et al. [20] drew attention to the strain on the hands during the subsequent phases of fruit picking, including manual transport to a basket and then manual transfer of that basket, for example, to a crate. Ng et al. [21] outlined the impact of a fruit picker's non-ergonomic body position during fruit picking on work quality, particularly on the quality of biological materials. In this context, it was highlighted how improper working conditions lead to musculoskeletal pain and disorders. The study found that individuals with permanent MSDs and those exposed to poor ergonomic conditions were almost 50% less efficient during fruit harvest compared to perfectly healthy individuals [22]. The evidence came from a study by Jara et al. [23]), which employed the surface electromyography (EMG) method to determine the level of electrical potential in muscles during manual effort, thus assessing the dynamic load on muscles]. A similar study was conducted by Komarnicki and Kuta [14], who used the EMG method to measure the load levels of individual musculoskeletal segments and their impact on the quality of harvested strawberries.

Thetkathuek et al. [18] highlighted the emergence of significant loads, particularly in the cervical and lumbar areas of the spine, during load testing. They also demonstrated, through numerous studies, that women are a more vulnerable group when it comes to acquiring musculoskeletal disorders (MSDs). However, in their opinion, further studies and diagnostics of working conditions are needed to determine the additional consequences of working in non-ergonomic conditions. In a study conducted by Kim et al. [15], the authors noted that during their tests, more than 60% of fruit pickers suffered from upper limb disorders. As recommended by the authors, it is necessary not only to address the issue of ergonomics but also to implement solutions aimed at reducing the strain on workers during harvesting. A particularly interesting study was carried out by Li et al. [24], who conducted a comprehensive analysis of tomato harvesting techniques and identified the most crucial factors that have a significant impact on the physical strain experienced by the picker. By implementing a mathematical model, they described the relationships that exist between various factors involved in tomato harvesting work. Considering the significant

problem of the physical strain on pickers during the harvesting of both vegetables and fruits, efforts have been made to introduce modern solutions. In cases where there has been no innovation in fruit or vegetable picking, rules have been developed for the proper harvesting of fruits or vegetables. These rules take into account the positioning of different segments of the picker's body in relation to each other, the correct position for observing the object on the stem, and coordination between the described parameters [25]. The changes have contributed to improved efficiency in the manual harvesting of fruits and vegetables.

As a result, the purpose of this study was to evaluate the impact of harvesting techniques on picker ergonomics and the preservation of tomato quality. These considerations have led to the identification of two different concepts in this study. The first concept suggests that the position of the picker's body affects the level of loads, surface pressures, and contact area generated during the contact of the hand with the harvested tomato. Another approach assumes that the position of the picker's body during tomato harvesting directly influences the level of strain on their musculoskeletal system. This research is important in horticultural practice, in which ergonomically correct manual harvesting will ensure high quality of the fruit. From the picker's point of view, it will be possible to ensure proper work conditions which will reduce the risk of disorders in their musculoskeletal system.

## 2. Materials and Methods

### 2.1. Greenhouse Tests—Grip System Pressure Tests

In the first phase, greenhouse tests were conducted, during which tomatoes of the '*Buenarosa*' were manually harvested at the Guszpit horticultural farm located in Nowa Wieś Wrocławska, Poland. The fruits were grown in a hydroponic greenhouse on rock wool substrates (Plantop NG2.0 cubes and Grodan GT Master mat). The tomatoes were planted in February 2022, and the vegetation time was stable, air-conditioned, with daylight. The plants developed on nutrients with a 5 pH and an N:K ratio of 1:1.25. The fruit classified as round were harvested in July 2022 at a bright red maturity, at a temperature of ($20 \pm 0.2$ °C), with a relative humidity of 71%. The experiments involved simultaneously measuring the surface pressures applied by the right-hand fingers and recording the muscle tensions of the picker. The tests of human pressure applied during tomato picking were conducted using a Tekscan system (South Boston, MA, USA). The key component of the system was a foil sensor (Grip Sensor model 4256E) attached to anatomically important zones of the picker's hands and fingers (Figure 1).

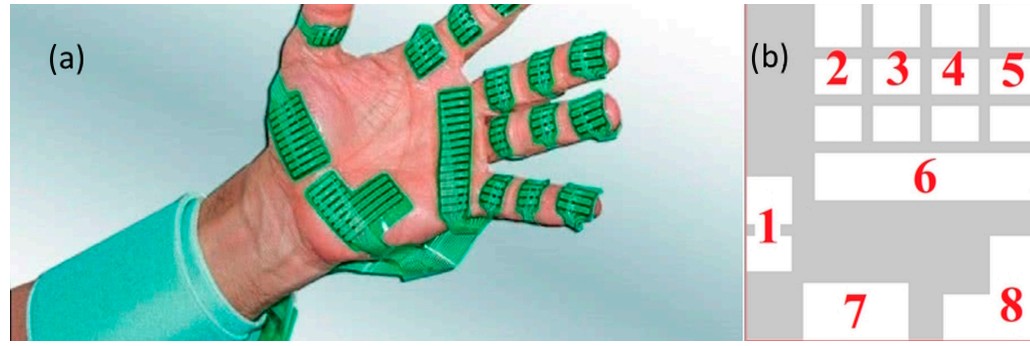

**Figure 1.** Picker's hand equipped with Grip System local pressure point identification sensor (**a**), together with the marking of the measurement zones of the right hand; (**b**) (1—thumb, 2—index finger, 3—middle finger, 4—ring finger, 5—little finger, 6—muscle under the fingers, 7—thenar eminence muscle, 8—hypothenar eminence muscle).

The pressure sensor used had 18 working areas, a pressure range of 0.345 MPa, a thickness of 0.1 mm, and a sense density of 6.2 sens×cm$^{-2}$. Data transmission to the computer was conducted via a hub (VersaTek 2–Port Hub) connected to a grip (VersaTek Cuff), inside which a pressure sensor was positioned. The system, in conjunction with F–Scan Research software, enabled real-time data recording at sampling rates of up to

750 Hz. Greenhouse measurements were conducted in five repetitions for each of the three harvesting positions (1—spinal inclination (SI), 2—spinal twist (ST), and 3—wrist twist (WT)). The duration for a single measurement was 60 s, during which information was collected from 10 to 15 fruits. Approximately 220 tomato fruits were harvested during the tests. The measurement system displayed surface pressure distributions based on the recorded loads and contact surfaces. The stratified images analyzed in the tests pertained to the phases of maximum surface pressure occurrence.

### 2.2. Surface Electromyography

For the ergonomic area, the surface electromyography system (EMG) from NORAXON (Scottsdale, AZ, USA) was utilized. It is a modern, non-invasive method for assessing muscle tension values and maximum muscle strength during manual tasks. The device holds international certifications from SENIAM and ISEK for measurement accuracy. It comprises software on a portable computer, four wireless sensors (a four-channel device), and gel electrodes that are non-invasively attached to the skin of the test subject. Before conducting measurements, the test's subject skin is thoroughly cleaned to eliminate potential impurities. EMG data can be used to determine the forces generated during muscle activity throughout various phases of an activity. The sampling frequency was set at 1000 results per second. Electromyography can be used to evaluate the tension of all human muscle groups. The measurement error of the device is 2 mV. For example, the average range of muscle static potential is −40 mV to +40 mV during muscle engagement in light work. In addition to analyzing spatiotemporal parameters and kinematic quantities, surface electromyography is used to determine the correct posture during physical activity. EMG analysis accurately describes muscle tension parameters, making it a more precise measure than previously employed methods like heart rate assessment. The EMG system consists of surface electrodes attached directly to the subject's skin, preamplifiers, a Wi-Fi receiver, and a computer system (Figure 2).

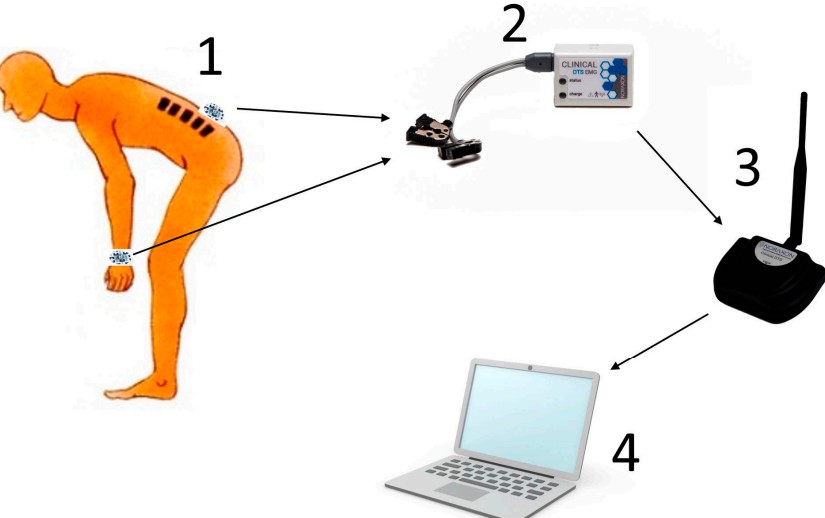

**Figure 2.** Diagram of EMG system. 1—electrodes; 2—preamplifier; 3—Wi–Fi adapter; 4—computer. Own source.

### 2.3. Examined Pickers

The tests were carried out under actual conditions during tomato harvesting. Based on observations, two muscle groups directly involved in the fruit harvesting process were selected. These muscles are responsible for coordinating hand movements and are strained during twisting and body inclining, specifically the muscles of the lumbar spine (Figure 3).

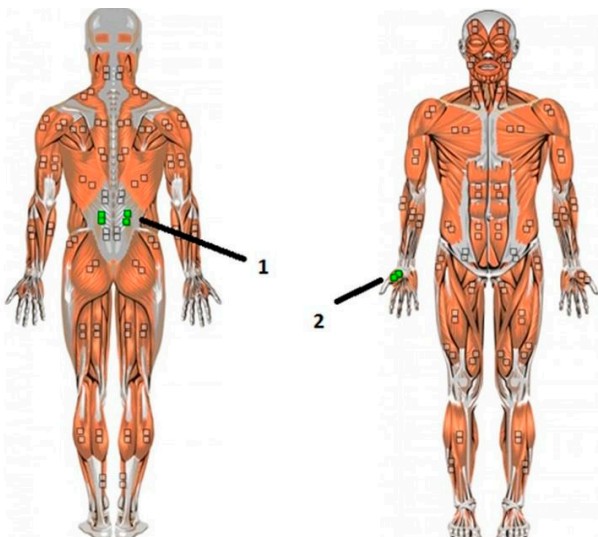

**Figure 3.** Tomato picker muscle groups examined during harvesting. Designations; 1—lumbar muscle (right lumbar—RT and left lumbar—LT); 2—right abductor pollicis brevis muscle.

Three characteristic body positions of the picker during tomato harvesting were determined for the tests and shown in the following Figure 4a–c.

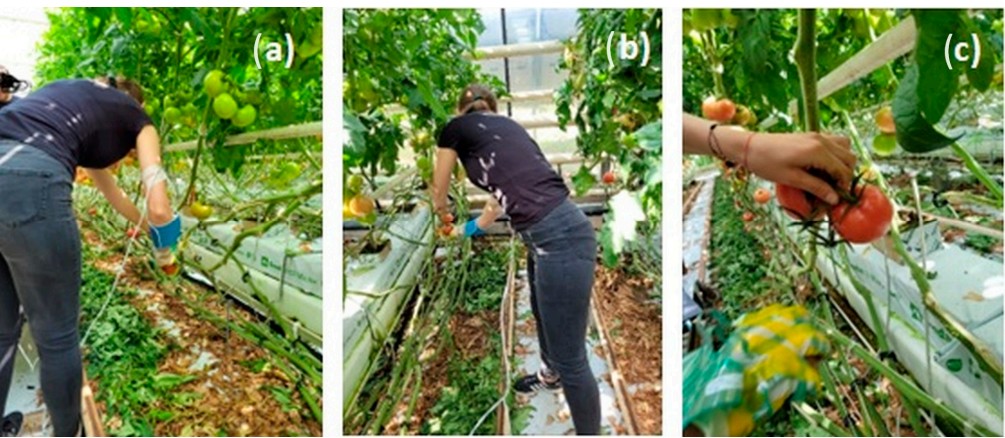

**Figure 4.** Characteristic body positions of a picker during tomato harvesting and designated levels of physical load; (**a**)—Load on the picker's hand vs. the spinal inclination over time; (**b**)—Load on the picker's hand depending on the torso twist over time; (**c**)—Load on the picker's hand depending on the wrist twist over time.

*2.4. Laboratory Tests—Compression Tests*

In the second phase of the tests, the tomato fruits were transported to the agrophysics laboratory of the Institute of Agricultural Engineering, where the material was immediately selected in terms of geometry and weight, and the firmness of 60 selected tomatoes was tested. The laboratory temperature was ($28 \pm 1$ °C) and the relative humidity was 52%. The weight of a single fruit was determined using an electronic balance (Steinberg, SBS–TW-500, Hamburg, Germany) with a range of 500 g and an accuracy of 0.01 g. The average diameter was determined using an electronic caliper with an accuracy of 0.01 mm (Hogetex, Varsseveld, The Netherlands). To assess the harvest maturity of the tomato flesh, firmness tests were conducted using an electronic firmness meter (Digital fruit firmness penetrometer, GY–4, by Newtry, Huizhou, China), with an accuracy of 0.01 N, 8 mm pivot diameter. The penetrometer was mounted on a high-precision manual tripod with digital length measurement (Sauter, TVL, Freiburg im Breisgau, Germany), ensuring repeatable sliding conditions at similar speeds and loads. Whole-fruit compression tests were performed on a

selected group of tomatoes using an Instron 5566 testing machine (Norwood, MA, USA) integrated with the Tekscan surface pressure system [13]. The fruit was placed in a lateral position and subjected to compression until destruction on a non-deformable substrate, with a constant head displacement speed of 10 mm/min. The pressure sensor (model 5076, I–Scan system) was placed under the fruit (Figure 5).

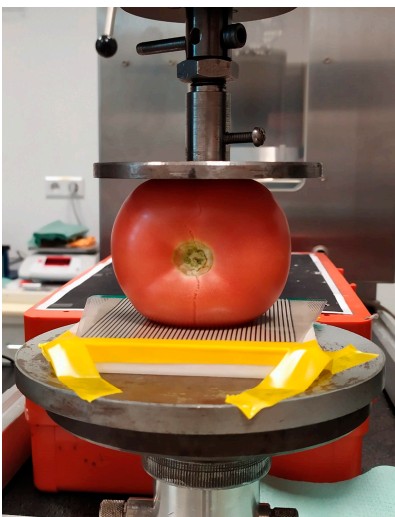

**Figure 5.** Compression test integrated with surface pressure measurement.

The synchronized apparatus had a sampling frequency of 0.1 s. Compression tests were conducted in 3 series, with 15 fruits tested for each harvesting position, totaling 45 fruits used. The above tests made it possible to simultaneously measure failure loads, deformations, contact surfaces, and maximum surface pressures.

*2.5. Statistics*

The data obtained were subjected to statistical analysis using Microsoft Excel and STATISTICA 12 (StatSoft Polska Sp. z o.o., Kraków, Poland). The basis for the statistical study was the evaluation of the relationship between the body position assumed by the picker during tomato harvesting and the contact surface, surface pressure, and force generated on the picked fruit. Another aspect of the statistical analysis was the assessment of the impact of the body position during harvesting on the level of the generated electrical muscle tension in the lumbar muscles and right wrist. To assess the variability of the results across the three body positions, the ANOVA and Student's t-tests were performed. In the analysis, the differences were considered statistically significant when $p < 0.05$ (*p*-value probability). In addition, basic statistics (mean values, deviations, and standard errors) were also calculated.

**3. Results**

*3.1. Greenhouse Test Results—Pressure Measurements*

From the perspective of the quality of the obtained material, the mechanical interactions between the hand and the tomato fruit were crucial. Figure 6 shows the overall changes in maximum loads and surface pressures generated by the picker's hand in different harvesting positions. The results indicated a significant effect of harvesting position on changes in loads and surface pressures ($p < 0.05$) occurring during tomato fruit picking especially in positions 1 and 3. Harvesting in position 3 (WT), which yielded the lowest surface pressures of 103 kPa, proved to be the most comfortable for the picker and the least taxing on the fruit. For position 1 (SI), the highest load values of about 33 N and pressures of 114.5 kPa were recorded. This could be attributed to the increased depth of body inclination compared to the other positions. The elevated parameter values in position 1 (SI) indicate the difficulty in picking fruit.

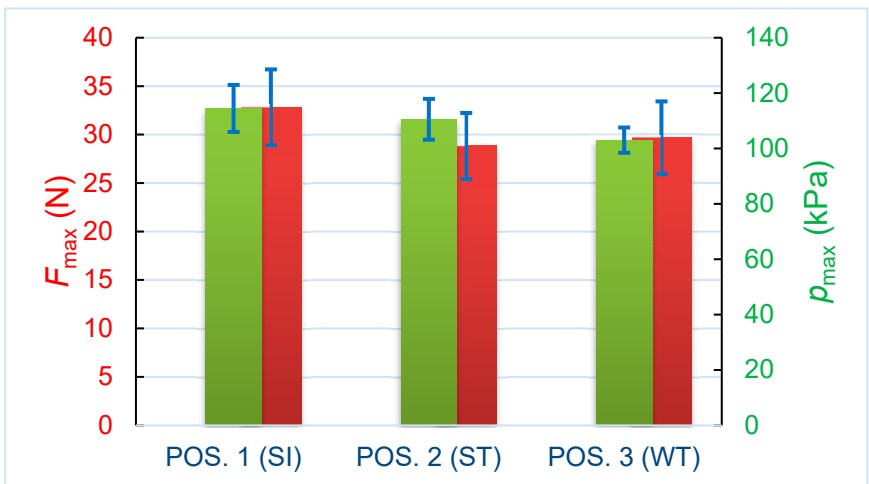

**Figure 6.** Changes in maximum loads and surface pressures recorded by the Grip System in different harvesting positions (Position 1—Spinal Inclination (SI), Position 2—Spinal Twist (ST), Position 3—Wrist Twist (WT)). Error bars indicate mean $\pm$ SD.

The tests also analyzed the variation in the contact area of the hand depending on the working position and its impact on picking time (Figure 7). The results indicate a statistically significant effect ($p < 0.05$) of changes in the contact area of the picker's hand depending on the harvest position. In position number 1 (SI), in addition to increasing loads (Figure 6), there was an increase in the contact area between the hand and the fruit being picked, reaching 2231 mm$^2$ (Figure 7). In the last comfortable hand position (position 3, WT), the angle of the wrist twist reduced the stability of the hand position and increased the tomato picking time to 2.26 s, compared to the other body positions during harvesting.

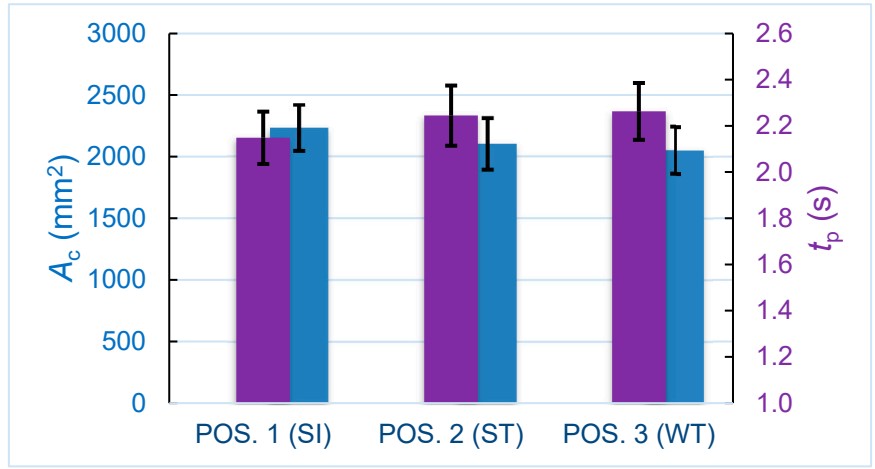

**Figure 7.** Changes in the maximum contact areas of the picker's hand and the time taken to pick the fruit recorded by the Grip System at different picking positions (Position 1—Spinal Inclination (SI), Position 2—Spinal Twist (ST), Position 3—Wrist Twist (WT)). Error bars indicate mean $\pm$ SD.

The Grip measurement system made it possible to observe the repetitive nature of the course of force pulses and the contact area of the hand fingers as a function of time. Three phases were distinguished in the process of picking a single tomato fruit. In the example force pulse (Figure 8), there was initially a rapid increase in value to a maximum, associated with grasping and ripping the fruit from the shoot (twisting the fruit until the stem breaks off from the shoot). In the second phase, after the fruit was ripped off, the hand was relieved, leading to a decrease in the force value by about 50–60%, until reaching a certain stabilized level. At this point, a third phase of the process occurred, which involved

holding and putting the fruit into the container. These processes were similar across all three adopted body positions.

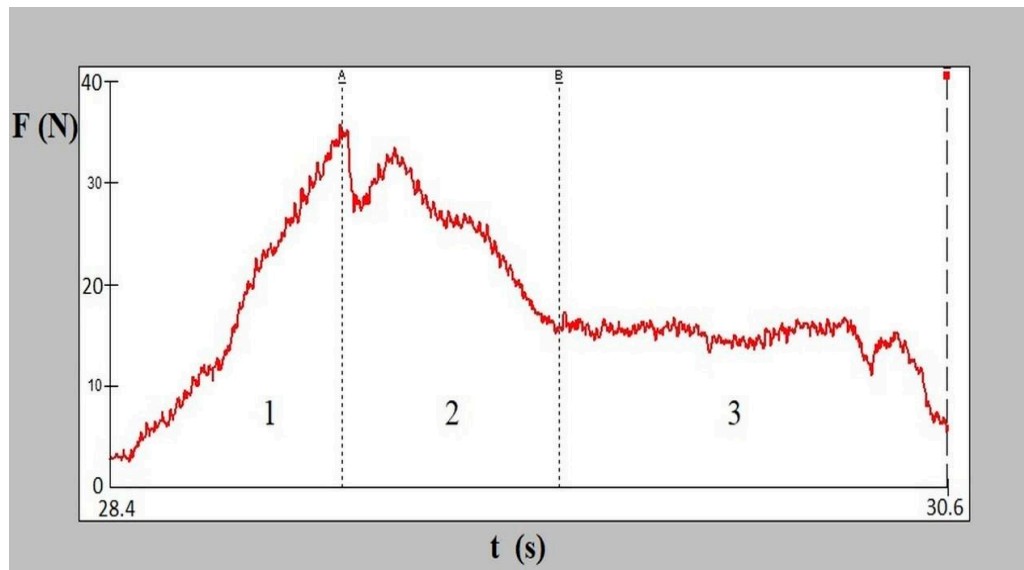

**Figure 8.** An example of the time course of a load pulse recorded during the fruit picking process (process phases: 1—ripping off, 2—relief after ripping off, 3—holding on). A and B are the boundaries that separate process phases.

Figure 9 shows percentages of surface pressure on each finger of the picker's hand and their stratified images formed during contact with the picked fruit in three working positions. The tests revealed that zones 6, 7, and 8 were minimally involved in pressure generation, so they were excluded from the analysis. It was observed that, in practice, the thumb was consistently the most heavily loaded in every position (27–36% of the entire hand), while the middle finger was rarely engaged in picking (1–5% of the entire hand). In position number 1, the primary fingers involved were (1), (5), and (2), which accounted for 36%, 32%, and 25% of the generated pressure for the whole hand (Figure 9a). In position 2, it was noted that the contribution to picking of the ring finger (4) increased to 10% (Figure 9b), and the wrist twist in position 3 increased the wrist's contribution to 22%, thereby equalizing the proportion of pressure among the four fingers of the picker's hand, ranging from 22% to 27% (Figure 9c). The tests showed that harvesting in position 1 primarily engaged three fingers (1), (5), and (2), while in positions 2 and 3, an increase in the contribution of the finger (4) during picking resulted in relieving the workload of the other fingers).

*3.2. Ergonomic Analysis*

This section presents the results of the loads generated during different tomato harvesting techniques.

3.2.1. Position 1: Load of the Picker's Hand vs. Spinal Inclination

Figure 10 shows the position of hands while picking tomatoes. In this position, it is evident that there is an obtuse angle at the elbow joint. Figure 11 also depicts the typical position of the lumbar spine, which often forms a right angle.

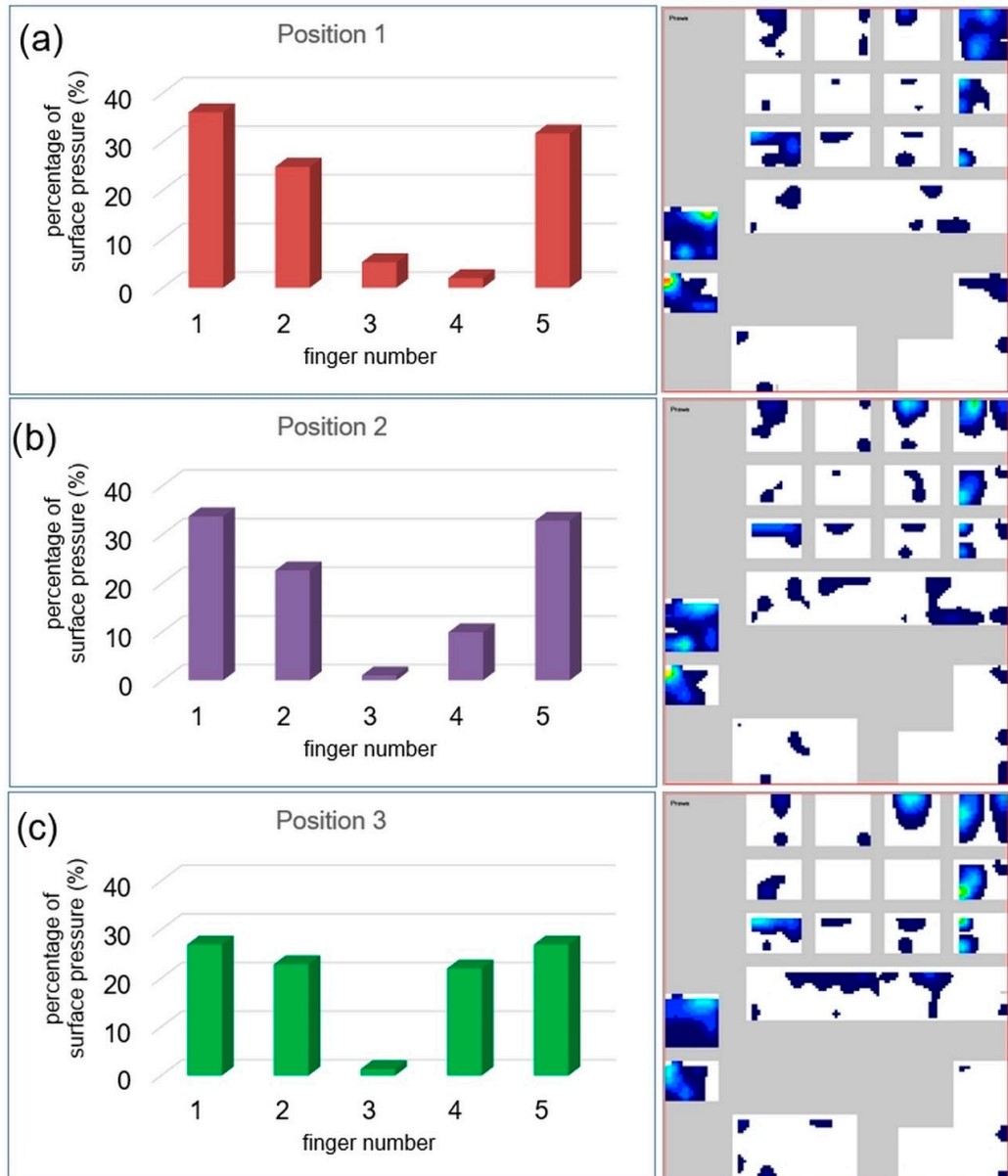

**Figure 9.** Percentages of surface pressures of individual fingers of the picker's hand and stratified images occurring during picking in the body positions assumed (**a–c**).

Figure 12 displays the relationship between normalized EMG signals (% MVC) of a tomato picker's hand and spinal inclination. The analysis reveals that the position of the spine is crucial, as forward bending alters the center of gravity of individual segments in the musculoskeletal system. Furthermore, reaching for the fruit requires precision. From an ergonomic standpoint, the spine's position is determined by the visibility of the fruit on the stem. When the fruit is lower, the picker's head is lowered, leading to increased muscle tension in the lumbar region. In such a body posture, static muscle loading predominates, often resulting in pain and tingling. The observations were divided into three phases. The first phase involves initiating contact with the tomato, the second phase entails grasping the tomato in hand and gently detaching it from the stem, and the third phase encompasses the detachment and manual transportation of the tomato. In the highest position, it reached 25% MVC at a spinal inclination of 50°, and as the level of inclination decreases, the muscle motor action signal also decreases.

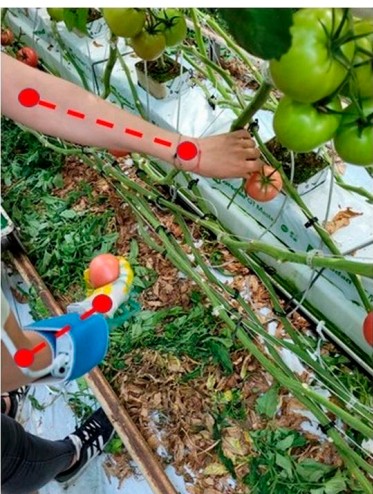

**Figure 10.** Typical hand positions during tomato harvesting. Own source.

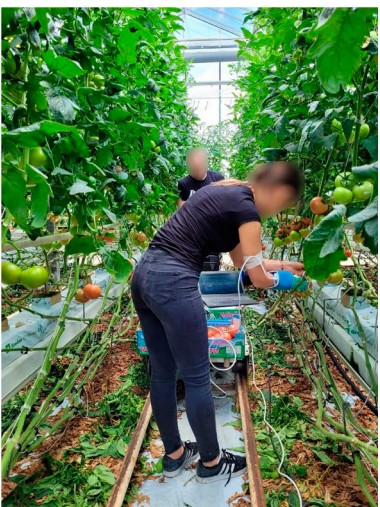

**Figure 11.** The inclination of the body in the lumbar part during tomato harvesting. Own source.

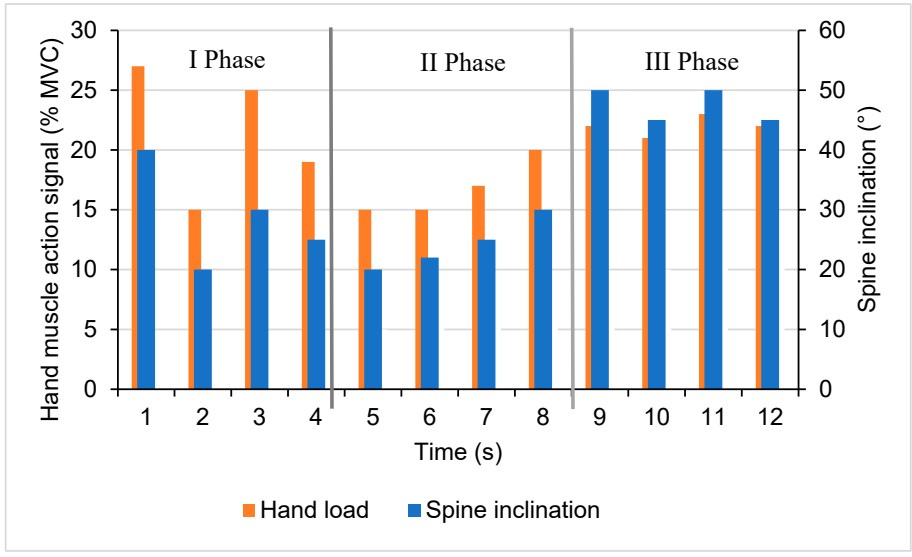

**Figure 12.** Selected course of hand loading over time depending on the angle of spinal inclination.

### 3.2.2. Position 2. The Picker's Hand Load Depending on the Torso Twist over Time

Figures 13 and 14 show the characteristic positioning of the body, including the torso twist, during tomato harvesting.

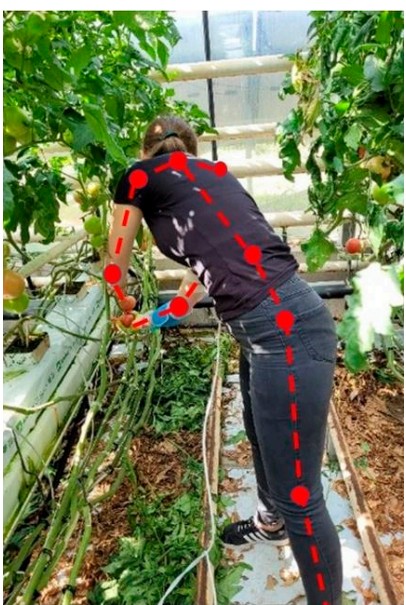

**Figure 13.** The characteristic torso twist during tomato harvesting that accompanies the body position inclination. Own source.

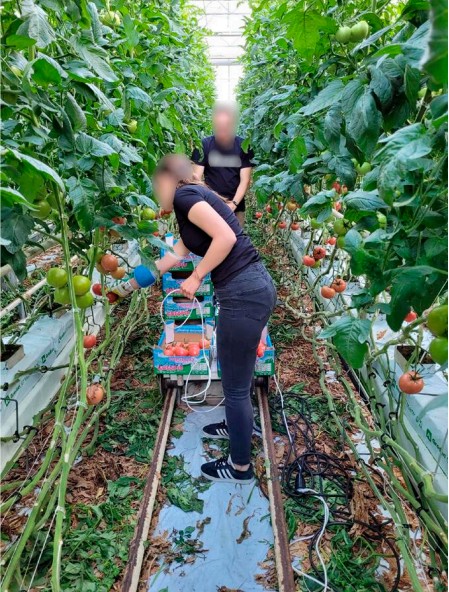

**Figure 14.** Body inclination in the lumbar part during tomato harvesting. Own source.

Figure 15 depicts the relationship between hand loading and the angle of torso twist. From an ergonomic perspective, this is a highly demanding position because the level of muscle tension increases significantly. The position of the spine (back) is unnatural and not parallel to the position of the legs, resulting in discomfort and pain in the pelvic area of the hip. The extent of discomfort and pain depends on the degree of angular twist. In this study, the highest values reached a 28% MVC level, with the largest twist angle measuring 85°. In most common cases, the mass of the lifted object also plays a crucial role and influences the level of muscle tension.

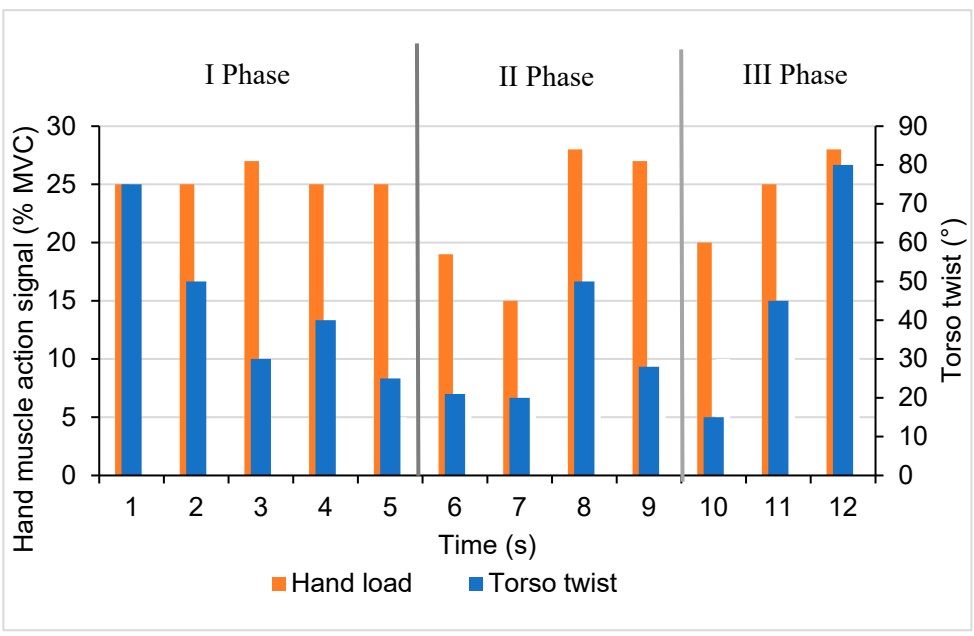

**Figure 15.** Selected course of the picker's hand load depending on the torso twist over time.

The lowest average load level occurs during phase number 2.

### 3.2.3. Position 3. The Picker's Hand Load Depending on the Wrist Twist over Time

In this position, the analysis primarily focused on wrist twists and their impact on hand loading. Figure 16 schematically shows how angular changes in the wrist can be analyzed.

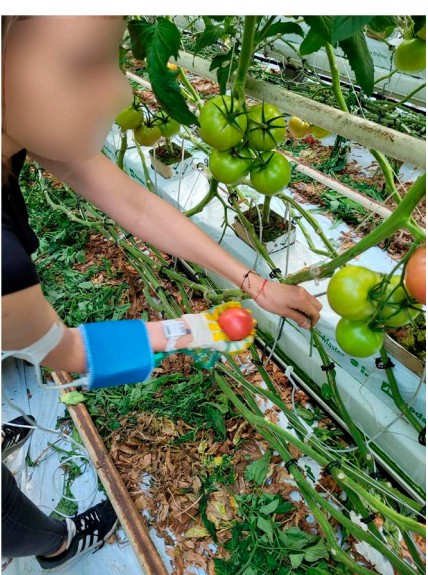

**Figure 16.** Position of the hand as it grasps the tomato from the stem. Own source.

Figure 17 demonstrates the relationship between hand loading and wrist twist angle. The highest muscle motor unit action potential values reach a level of 28% MVC and are associated with a wrist twist angle close to 36°. Excessive loading often leads to wrist joint irritation, and prolonged strain on the joints is one of the most common causes of disorders. Symptoms may include redness, swelling, and pain when resting or moving. A twisted wrist increases muscle tension. Injuries such as joint instability, accelerated degenerative change, and chronic wrist pain are quite common. The magnitude of this tension (and consequent load) primarily depends on the position of the fruit on the stem, its size, mass,

and shape. Analyzing the figure shows that the average picker's hand load occurs during the first and second phases of harvesting. This is attributed, among other factors, to the formation of convenient access to the grasped fruit.

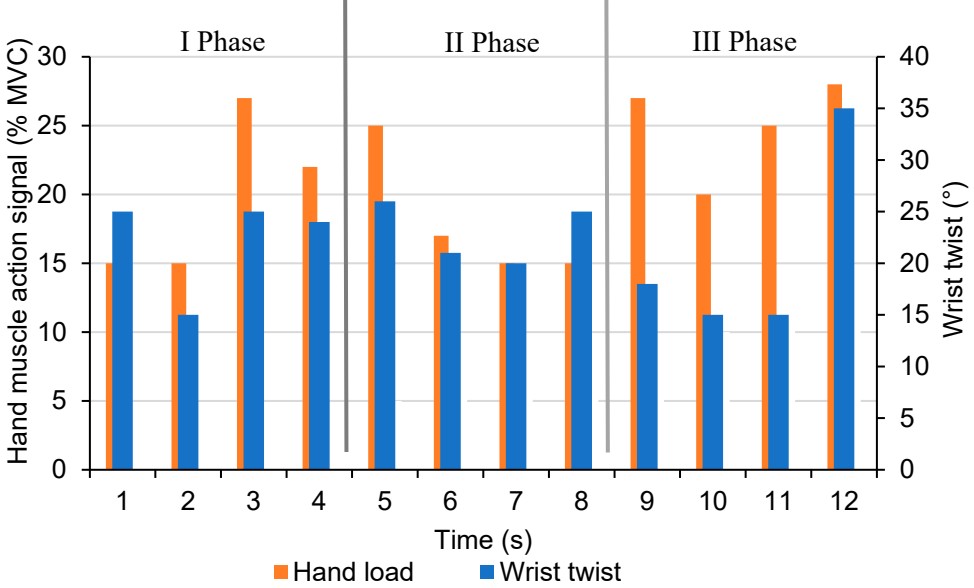

**Figure 17.** Selected course of the picker's hand load depending on the angle of wrist twist.

### 3.3. Laboratory Test Results—Destructive Compression Tests

In the second phase of tests, the material was selected with an average weight of $187.4 \pm 6.3$ g, an average diameter of $73.9 \pm 1.2$ mm and a firmness of $17.9 \pm 2.5$ N. Statistical analysis of whole-fruit compression tests showed no significant differences in recorded destructive loads, contact areas as well as surface pressures for fruits harvested using different methods ($p > 0.05$). The ANOVA analysis showed a probability ($p$) level of $p = 0.06$. This suggests that the work position during harvesting has no effect on changes in the strength parameters of the fruit—the quality of the material. Tomato compression tests performed with the I–Scan apparatus, on the other hand, showed an overall increase of 41% on average in destructive loads compared to Grip System measurements, which averaged 32% in destructive pressures (Table 1).

**Table 1.** Comparison of loads, contact areas, and surface pressures generated by hand during fruit picking (Grip–System measurement) with values recorded during destructive compression tests (I–Scan measurement).

| Harvest Position | Grip System | | | I–Scan | | |
|---|---|---|---|---|---|---|
| | $F_{max}$ | $A_c$ | $p_{max}$ | $F_{max}$ | $A_c$ | $p_{max}$ |
| | (N) | (mm$^2$) | (kPa) | (N) | (mm$^2$) | (kPa) |
| 1 (SI) | $32.8 \pm 3.9$ | $2231.4 \pm 187.2$ | $114.4 \pm 8.4$ | $45.8 \pm 7.2$ | $936.3 \pm 128.8$ | $147.9 \pm 23.6$ |
| 2 (ST) | $28.8 \pm 3.4$ | $2101.2 \pm 208.6$ | $110.5 \pm 7.3$ | $42.5 \pm 7.1$ | $942.7 \pm 103.6$ | $138.6 \pm 24.7$ |
| 3 (WT) | $29.6 \pm 3.7$ | $2050 \pm 190.5$ | $103 \pm 4.5$ | $42 \pm 7.2$ | $899.5 \pm 153.5$ | $146.4 \pm 19.4$ |

Data are presented as mean $\pm$ SD.

Regarding destructive compression tests, the verification of loads and surface pressures measured by the Grip System indicates that tomatoes harvested in three different positions were not subject to damage.

Figure 18 displays images of peak surface pressures measured during the tomato compression process. Initially, the maximum surface pressures are point-like, concentrating in the center of the contact surface (Figure 18a). Subsequently, due to the soft seed chambers

(locular cavity) and filling of free cellular spaces, pericarp deformation occurs, resulting in an outward displacement of loaded zones while relieving central zones (as illustrated in Figure 18b,c at 30 s and 60 s into the process). Further increases in displacement lead to peak surface pressures (Figure 18d), followed by skin damage and a sharp drop in value. The mechanical resistance of the tomato fruit is primarily attributed to its relatively tough skin and the outer wall of the pericarp.

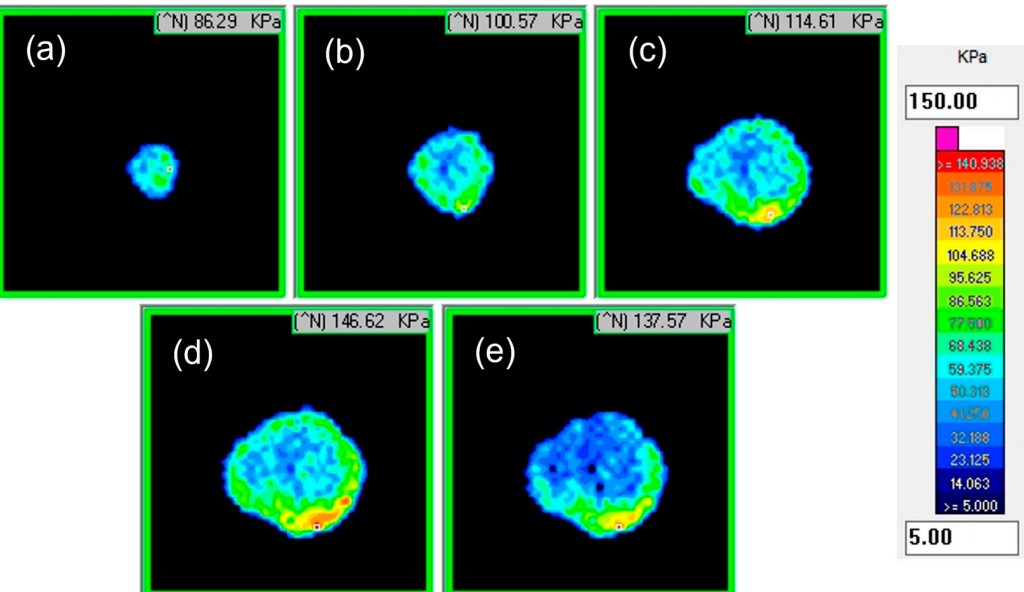

**Figure 18.** Stratified images of surface pressures recorded during tomato fruit compression test: (**a**)—contact-type pressure at 5 s, (**b**,**c**)—outward displacement of loaded zones, with central zones relieved at 30 s and 60 s, (**d**)—reaching a local maximum at 82 s, (**e**)—decrease in pressure after fruit damage at 85 s.

## 4. Discussion

This paper presents an innovative research concept that aligns fruit quality with the quality of human work during manual harvesting. In pursuit of this concept, a measurement system for surface pressure on the fruit, known as the Grip System, and a system for recording electric potential (EMG) in muscles were employed. The sensor-equipped glove offers the opportunity to assess the pressure value, which increases as the picker touches the fruit at various points inside the hand. The recorded signal was transmitted to a computer system and saved. In today's consumer-centric market, product quality is of paramount importance, and to ensure high quality, all negative factors must be mitigated at each stage of the production process, including during the harvest phase.

This type of damage can be prevented through proper ergonomic conditions during manual harvesting, such as reducing bodily strain, altering work techniques, and selecting the most suitable methods. To diagnose and implement these changes effectively, the right diagnostic technique is crucial. In this case, surface electromyography is the preferred method, as it can be employed under various conditions and types of work. Therefore, the authors utilized it to evaluate the electrical potential of muscles during manual tomato harvesting.

Most mechanical actions affecting tomatoes occur during harvesting and transportation, which can result in quality deterioration. Several authors analyzed similar issues earlier, paying attention to the relationships between biological material and ergonomics [13]. Komarnicki and Kuta demonstrated in their study, that during strawberry picking, body position matters and affects the quality of the harvested fruit, hence they used some inspirations and relationships in this paper. It was found that mechanical damage to the fruit is considered a defect in the biomaterial and is closely linked to the fruit's anatomi-

cal structure and mechanical interactions [26,27]. One such factor is the moment of fruit harvest and the right technique. Compression causes bruises and cracks in tomato fruit. Compression damage also occurs during mechanical picking if the gripping forces exceed the threshold for tissue damage. Compression tests have shown that the degree of compression, the curvature of the finger surfaces, and internal structural features affect the mechanical damage of tomato fruit [28–33]. In this study, the authors used a state-of-the-art method for evaluating the strength properties of tomatoes, the Grip System, based on direct measurement of the pressure of the hand fingers on the surface of the fruit in the range of loads encountered when picking a tomato. It was shown that during the picking of '*Buenarosa*' tomato at the bright red stage for the three tested body positions, the level of loads and surface pressures of the hand on the fruit did not exceed the critical compression values (about 42–45 N and 138–146 kPa) causing tissue damage. At present, there are few similar experiments to compare the results obtained for tomato fruits picked directly by humans. Available reports for robotic harvesting, in which ripping-off experiments were performed by finger effectors together with manipulators, indicate long picking times and relatively low harvesting efficiency [34,35].

Through numerical simulation, Zu et al. [36] presented the internal stress distribution of tomatoes, as well as the structural optimization design of a non-destructive post-harvest device, in which the peak contact stress was 107 kPa, close to the maximum pressures recorded by the Grip System. The Grip System apparatus showed that tomato picking by the picker took between 2 and 2.5 s. Similar results regarding harvesting time and picking forces were obtained by Gao et al. [5] in their study of a dynamic measurement system for manual picking of cherry tomatoes. They found that peak forces applied to the fruit during twisting and pulling ranged from 27 to 34.5 N, with the index finger contributing the most, followed by the thumb and middle finger. They also measured that the average maximum clamping force the fruit could withstand was 41.6 N. During the manual harvesting of tomato fruit from the plant, when a person's fingers grasp the target fruit, they tend to employ pulling, twisting, or bending methods to separate the fruit from the stem. The effect of finger tissue mechanics and structure on the mechanical characteristics of soft contact was investigated using finite element analysis by Hou et al. [37]. So far, there has been no clear progress regarding the quantitative impact of the mechanics and structural size of different human finger tissues on the mechanical behavior of various finger areas during grasping. In contrast to hard contact (e.g., apples, pears), the mechanical behavior of soft-contact fingers indicates that the contact force between the fingers and the fruit does not increase sharply with the deformation of the fingers when grasping tomato fruit, whose tissue structure is more flexible. Kuta et al. [38] proved that for apples, picking height affects the level of picker muscle tension and the surface pressure applied by the hand on the fruit. As the harvest height increased, the values of the average pressures generated by the thumb finger resulting from adopting an uncomfortable body position also increased.

The subject of the conducted tests was the measurement of musculoskeletal load on tomato pickers using a novel method of surface electromyography. The methods that have been used for ergonomic evaluation so far typically rely on point scales that provide an overall assessment of ergonomic load. Electromyography is an innovative method known for generating highly reliable test results, and it is used to analyze musculoskeletal loads [39]. However, manual harvesting techniques are commonly employed, especially when the quality of the fruits and vegetables picked is of utmost importance. To provide a comparative example, consider the tested load on farmers' hands using the EMG method during milk extraction with the bubble method. In such cases, the load during the attachment of the teat cups averaged 30% MVC for the 3 kg milking apparatus carried [40]. Nevertheless, there has been no in-depth analysis of forearm load that takes into account factors like the alignment of individual hand segments, which can produce acceptable EMG values for evaluation, and the maximum frequency of hand movement repetitions. The values of hand loading expressed in mV during tomato picking in the present study were comparable to the results of studies involving fish processing workers who performed

tasks such as lifting and ranged from 5 to 50 mV [41]. Musculoskeletal load tests in the lumbar region were also conducted using EMG among welders. Based on this, the load on this part of the spine was estimated to be 40% MVC. Exoskeletons were used to enhance work comfort and reduce muscle tension by about 25%. Typical signal amplitudes (peak-peak) for surface electromyography measurements reach up to 10 mV, with typical filters ranging from 1 Hz to 1 kHz and a sampling frequency of up to 5 Hz [42]. During these studies, several issues with the EMG system were noticed. In some cases, there were problems with transmitting the EMG signal from the preamplifier (signal sender) to the Wi-Fi adapter (signal receiver), typically due to the large distance between the devices, exceeding 20 m. Consequently, measurements needed to be closely monitored because a lack of communication between these devices would halt the measurement process. In such situations, the devices were placed closer to each other, and the authors also utilized a wired connection. The quality of the measurement also depended on the electrodes used and the degree of skin cleaning before measurement, as the signal needed to be undisturbed. In a few instances, perspiration-covered skin resulted in incorrect measurements. However, a significant advantage of the devices used in this study was their versatility, small size, and mobility, allowing them to be used effectively in various scenarios and conditions.

## 5. Conclusions

The purpose of this study was to evaluate the impact of harvest techniques in terms of picker ergonomics and preservation on tomato quality. The tests showed the significance of the effect of the picker's adopted body position on changes in maximum hand loads and surface pressures of picked tomato fruit. Harvesting in position 3 (WT) proved to be the most comfortable for the picker and the least taxing on the fruit, likely due to the increased depth of body inclination compared to the other positions. In the least comfortable hand position (position 3, WT), the angle of the wrist twist reduced the stability of the hand position and increased the tomato picking time. The fruit-picking process analysis made it possible to observe the repetitive three-phase nature of the course of force pulses and the contact area of the hand fingers as a function of time. The use of the Grip system confirmed that in practically every position the thumb was the most loaded (27–36% of the entire hand), while the middle finger was practically not involved in picking (1–5% of the entire hand). The analysis shows that the highest levels of load in the lumbar spine occur when the torso is twisted while the worker's posture is inclined. Consequently, this can lead to an increased risk of ailments, tingling of the limbs, and pain as exposure to the uncomfortable body position of the picker increases.

The use of I–Scan surface pressure measurements allowed for tracking the fruit destruction process, understanding alterations in tomato tissue structure, and monitoring the concentration of surface pressures under compressive loading. These studies hold significance in greenhouse production as they evaluate the ergonomics of the picker's work and the quality of the harvested material. In the future, such research can be applied to soft, delicate fruits, particularly those sensitive to handling, such as raspberries, blueberries, and strawberries. The tests revealed no significant impact of the tomato picker's body position on the degradation of picked fruit quality. However, these body positions do significantly influence the level of physical load and work comfort.

The above studies are important in greenhouse production for evaluating the ergonomics of the picker's work as well as the quality of the harvested material. In the future, they can be used for soft, small fruits especially sensitive as raspberries, blueberries, and strawberries. In addition, this study can be used to design the right type of exoskeleton–external construction on the body that supports the picker on the basis of EMG results.

**Author Contributions:** Conceptualization, P.K. and Ł.K.; methodology, P.K. and Ł.K.; validation, P.K., Ł.K.;formal analysis, P.K., J.P. and Ł.K.; investigation, P.K., Ł.K. and K.Ł.; resources, P.K., Ł.K. and K.Ł.; writing— P.K., J.P. and Ł.K.; writing—review and editing, P.K. and Ł.K.; visualization,

P.K.; manuscript translation, Ł.K. All authors have read and agreed to the published version of the manuscript.

**Funding:** The APC is financed by Wrocław University of Environmental and Life Sciences. The research is co-financed under the Leading Reasarch Groups support project from the subsidy increased for the period 2020–2025 in the amount of 2% of the subsidy referred to Art. 387 (3) of the Law of 20 July 2018 on Higher Education and Science, obtained in 2019.

**Institutional Review Board Statement:** All subjects gave their informed consent for inclusion before they participated in the study. The study was conducted in accordance with the Declaration of Helsinki, and the protocol was approved by the Ethics Committee of Senate Research Ethics Committee at Wroclaw University of Health And Sport Sciences (Resolution 14.12.2015).

**Data Availability Statement:** Not applicable.

**Acknowledgments:** Many thanks to owner of the tomato plantation, Dominik Guszpit, who provided help during the greenhouse tests.

**Conflicts of Interest:** The authors declare that they have no known competing financial interests or personal relationships that could have appeared to influence the work reported in this paper.

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
