# Peer review of "Tomato Fruit Quality as Affected by Ergonomic Conditions While Manually Harvested"

_agriculture, doi:10.3390/agriculture13091831_

Round 1
Reviewer 1 Report
Nowadays, consumers are particularly concerned about the quality of their products and this study was carried out to ensure good quality fruit and vegetables. The authors combined food quality with ergonomics in the manual harvesting process. Specifically, the authors used surface electromyography to compare the musculoskeletal strain of pickers in a qualitative study of hand-picked tomatoes. To this end, tomato quality was assessed using a grip strength system by evaluating the effect of picking hand pressure (three different picker postures) on picking surface pressure. This paper (ISSN 2077-0472) seems to present new findings and innovative research ideas. Writing is cursory, paying attention to the abstract, introduction, discussion and conclusion sections. Literature is presented in the wrong format. Problems with paragraphing, please rephrase or combine. It appears to be acceptable if revised as suggested below.
In abstract
1. P31: In the summary section sEMG appears only once, so it is recommended not to use the abbreviation.
2. P34: The lack of research significance.
3. P36: After checking the writing format of this journal, the first keyword needs to be capitalized. Please check the following writing format carefully.
4. P36: If you delete the abbreviation of sMEG, modify the keyword's sMEG as well.
In introduction.
5. P38-72: This section quite long. It can be shortened to raise to questions and knowledge gap and thus the significance of this study.
6. P77-122: These two paragraph format is wrong, need to adopt first line indentation.
7. P73-76: I think this paragraph needs to be considered again, because it does not constitute a paragraph just because of previous research.
8. In introduction: The format of the cited references is incorrect, please check carefully. For example, "Młotek et al. (2015)" should be changed to "Młotek et al. [13] ".
9. P128-129: Shouldn't the last paragraph be combined with the last one? Why start a new paragraph?
In materials and methods
10. 2.1 lack of experimental design in detail. Such as, sowing time, ratio of substrate, illumination, etc.
11. P148-158: Pay attention to paragraph formatting
12. Is this the first time that fruit packing and ergonomics have been mentioned? If mentioned in others, please add relevant references in materials and methods to better support the methods of this study
In results
13. Please add ANOVA and improve the beauty of the figure.
In discussion
14. Are the conclusions reached in this research supported by previous results? If so, you need to add appropriate information in this section.
In conclusions
15. P502-505: Incorrect structure. The conclusion section should be Purpose, Test Results, Summary and Significance. However, the logic of this paragraph is to start with a summary, followed by implications and future perspectives. Please make adjustments carefully.
Author Response
REVIEWER I
Dear Reviewer, -
Thank you very much for your comments, I hope these suggestions will enhance the quality of this paper.
- P31: In the summary section sEMG appears only once, so it is recommended not to use the abbreviation.
Answer
The records have been changed – (line 32).
- P34: The lack of research significance.
Answer
This research is important in horticultural practice, in which ergonomically correct manual harvesting will ensure high quality of the fruit. From the picker's point of view, it will be possible to ensure them proper working conditions which will reduce the risk of disorders in their musculoskeletal system – (lines 129-133).
- P36: After checking the writing format of this journal, the first keyword needs to be capitalized. Please check the following writing format carefully.
Answer
The record has been changed –(line 39).
- P36: If you delete the abbreviation of sMEG, modify the keyword's sMEG as well.
Answer
The record has been changed –(line 39).
In introduction.
- P38-72: This section quite long. It can be shortened to raise to questions and knowledge gap and thus the significance of this study.
Answer
Thank you for your attention: This section has been shortened according to your recommendations (lines 69-75).
- P77-122: These two paragraph format is wrong, need to adopt first line indentation.
Answer
The records have been changed.
- P73-76: I think this paragraph needs to be considered again, because it does not constitute a paragraph just because of previous research.
Answer
This is part of earlier paragraph, I have matched these descriptions – (lines 66-79).
- In introduction: The format of the cited references is incorrect, please check carefully. For example, "Młotek et al. (2015)" should be changed to "Młotek et al. [13] ".
Answer
The records in the manuscript have been changed – (line 76).
- P128-129: Shouldn't the last paragraph be combined with the last one? Why start a new paragraph?
Answer
This sentences was matched with former part of manuscript (lines 127-129).
In materials and methods
- 2.1 lack of experimental design in detail. Such as, sowing time, ratio of substrate, illumination, etc.
Answer
Thank you for your comments. The details of tomato growing experiment have been completed in the
(lines 147-149).
- P148-158: Pay attention to paragraph formatting
Answer
A paragraph was reformatted – (lines 144-161).
- Is this the first time that fruit packing and ergonomics have been mentioned? If mentioned in others, please add relevant references in materials and methods to better support the methods of this study.
Answer
I have added our former paper - position number [14] - line 611. Position [13] after changes.
In results
- Please add ANOVA and improve the beauty of the figure.
Answer
Thank you for your comment. The readability of Figure 18 has been improved - (lines 403-404). The ANOVA result was added – line 383.
In discussion
- Are the conclusions reached in this research supported by previous results? If so, you need to add appropriate information in this section.
Answer
Several authors analyzed similar issues earlier, paying attention to the relationships between biological material and ergonomics [13]. Komarnicki and Kuta demonstrated in their study, that during strawberry picking, body position matters and affects quality of the harvested fruit, hence they used some inspirations and relationships in this paper. It was found that mechanical damage to the fruit is a defect in the biomaterial and is closely related to the anatomical structure and mechanical interaction of the fruit [27,28]. One of such factors is the moment of fruit harvest and right technique. Lines 427-434.
- P502-505: Incorrect structure. The conclusion section should be Purpose, Test Results, Summary and Significance. However, the logic of this paragraph is to start with a summary, followed by implications and future perspectives. Please make adjustments carefully.
Answer
Thank you for your comments. The conclusions have been redacted and arranged according to your recommendations (lines 510-549).
Language
Answer
The language throughout the manuscript has been extensively revised.
Reviewer 2 Report
l The last sentence in the abstract seemed unfinished. And what’s the results of the work?
l In the hand strength related test, how to select the experimenter? How to solve the problem of unequal power between men and women?
l In Figure 5, a single-direction tomato-related test was taken, is it the same as applying multi-directional force by the hand? I don’t think so.
l At the end of the article, ergonomic related advice was given, but everyone's subjective wishes are different, height and weight are also different, are these issues taken for consideration?
l Future work was not mentioned in the conclusion.
l Some references are too old.
l Some of the pictures in the article are not very clear and standardized, like figure 18.
Must be improved.
Author Response
REVIEWER II
Dear Reviewer,
Thank you very much for your comments, I hope these suggestions will enhance the quality of this paper.
- The last sentence in the abstract seemed unfinished. And what’s the results of the work?
Answer
Thank you for your comment. The summary has been finished as recommended - lines 35-38.
- In the hand strength related test, how to select the experimenter? How to solve the problem of unequal power between men and women?
Answer
In this case, level of strength depends on several factors. The basis are anthropometric features, age of the person, assessment of whether the work is monotonous, frequency of work during the day (number of repetitions) and the degree of muscle training (depending on the repeatability of the activity, length of service, and body position at work). Based on these criterias it is possible to select right person for the study, which is characterized by average parameters.
- In Figure 5, a single-direction tomato-related test was taken, is it the same as applying multi-directional force by the hand? I don’t think so.
Answer
Thank you for your comment. We agree that when harvesting the tomatoes by hand, forces in different directions occur in the places where the hands it touch. Authors decided to conduct the studies objectively, repeatably and precisely to determine the maximum destructive load in the popular uniaxial compression test (in the lateral position) in order to estimate the reference value of the force, which was compared to the forces (pressures) recorded during destructive in various positions. We think that conducting a multidirectional destructive test using a human hand would be impossible due to the variable load rate or the inability to determine the maximum destructive value (a human hand, due to its flexibility, will not allow observe the moment of damage to the fruit tissue).
- At the end of the article, ergonomic related advice was given, but everyone's subjective wishes are different, height and weight are also different, are these issues taken for consideration?
Answer
Of course, this is very important because each person has different anthropometric features, but these dimensions can be organized by creating, for example, classes of anthropometric features or generating solutions for pickers based on an anthropometric atlas divided into body size (height, body weight,, limb length and gender features). An optimal solution can be prepared for each respondent, which takes into account a listed factors.
- Future work was not mentioned in the conclusion.
Answer
Thank you for your comment. We have provided information regarding the possibility of use the above research in future work. In addition, this study can be used to design the right type of exoskeleton - external suport construction for pickers on the basis of EMG results – lines 545-550.
- Some references are too old.
Answer
We have removed the oldest literature position from the manuscript. Lines – 602-603; 664-665; 683-685.
- Some of the pictures in the article are not very clear and standardized, like figure 18.
Answer
Thank you for your comment. We have improved the readability of Figure 18. Lines – 403-404.
Language
Answer
The language throughout the manuscript has been extensively revised.
Round 2
Reviewer 1 Report
The entire manuscript has been carefully revised and approved for publication.
Reviewer 2 Report
No more.
No more.